# Dose-Related Effects of Endurance, Strength and Coordination Training on Executive Functions in School-Aged Children: A Systematic Review

**DOI:** 10.3390/children9111651

**Published:** 2022-10-28

**Authors:** Alina Drozdowska, Gernot Jendrusch, Petra Platen, Thomas Lücke, Mathilde Kersting, Kathrin Sinningen

**Affiliations:** 1Research Department of Child Nutrition, University Hospital of Pediatrics and Adolescent Medicine, St. Josef-Hospital, Ruhr University Bochum, 44791 Bochum, Germany; 2Department of Sports Medicine and Sports Nutrition, Ruhr University Bochum, 44801 Bochum, Germany

**Keywords:** physical activity, cognitive function, fitness, schoolchildren, training

## Abstract

This systematic review aims to evaluate previous findings on the dose-related effects of short- and long-term physical activities (PA) on executive functions (EF) using a new approach by considering the success of experimental manipulation. Eight electronic databases were searched between May 2021 and September 2021. Randomized control trials among healthy children (6–12 years) were screened. Data extraction included the measurement of experimental manipulations and pre–post measurements of physical fitness. After identifying 1774 records, 17 studies were included (nine short-term PA and eight long-term PA). The overall results suggest that a single 20-min PA may be overwhelming for short-term EF in children up to 9 years of age but may be beneficial for children 9 years and older. A dose-related relationship between PA and EF could not be verified in long-term studies, which is possibly due to insufficient fitness gains and participation in the intervention. Short- and long-term endurance and coordination training could improve children’s executive functions, but so far, there is no specific evidence on the duration, frequency, and intensity of PA. Not quantity but quality of intervention seems to be important in this context. Further intervention studies are needed that control for the characteristics of the experimental manipulation.

## 1. Introduction

Determining predictors of executive function, higher-order cognitive abilities, in children is important for learning and success throughout school life [1,2]. The causality that any physical activity (PA) has an impact on these abilities has not been established [3]. Although controversially discussed, there is agreement that for cognitive development, ‘the more PA the better’ [4,5]. However, how much is enough to make cognitive progress and what type of exercise is beneficial? The most comprehensive review of causality between various PAs and executive function suggests that endurance, strength, and coordination training have different effects on these cognitive abilities, and those with greater cognitive demands appear to be most favorable. Automated aerobic and strength exercises seem to contribute the least to executive function improvement compared to coordinative training methods [3]. However, type- and dose-dependent long-term PA interventions related to executive function are still rare, particularly in randomized controlled trials (RCT) with school-aged children before puberty [6,7,8].

The limited number of such studies may be one of the reasons why the World Health Organization (WHO) guidelines do not include specific recommendations for cognitively demanding PA. In 2020, the WHO reformulated the PA guidelines and recommends daily aerobic exercise in addition to strength exercises at least three times per week to promote children’s mental and physical health [6]. Furthermore, the recommendations emphasize the importance of variety and fun in age- and performance-appropriate PA. A gradual increase in the complexity and difficulty of the activities should be desirable over time. This approach of continuous PA challenges could be crucial for fitness and cognitive development, as studies show [9,10,11]. For example, the result of a non-RCT study showed that only 8–10 min of high-intensity exercise over 12 school days for four weeks was sufficient to improve working memory in children aged 8–12 years [9]. Another non-RCT with children aged 7–10 years showed that an 8-week challenging exercise program twice a week for 90 min improved visual–spatial working memory [10]. Improved visual–spatial abilities were also achieved in adolescent students after a 12-week coordinative intervention that occurred twice a week for 40 min each [11]. These studies also showed improved fitness performance within the experimental manipulation. Such data collections are sometimes missing in studies with prepubescent children [12,13] or when improvement in cognitive skills has been demonstrated after a PA intervention [14,15,16]. Therefore, it cannot be concluded whether the intervention effects on cognition were related to fitness gains or were influenced by other factors such as motivation and enjoyment [3]. Checking the experimental success, e.g., fitness gains, could clarify the results.

In contrast to the long-term interventions, implementing “more PA is better” in the short-term training session does not appear to yield short-term cognitive success. The researchers suggest that too long or too vigorous PA could negatively impact short-term cognitive challenge and even lead to short-term cognitive decline [17]. In school life and during short-term cognitive demands such as school exams, it may be important when and how much PA occurs in school. Recovery breaks after physical exertion could potentially provide a cognitive boost. Schmidt, et al., stated that after 45 min of physical education with coordinative sessions, there was no immediate effect on attentional performance, but there was 90 min after completing [18]. Studies in which physical exertion is measured and additionally linked to executive functions could provide more information about this suspected mediation effect. Like the long-term PA interventions, the short-term effects can only be meaningful if the experimental manipulation (e.g., monitoring the real-time heart rate or subjective effort) was successful [19]. To our knowledge, no systematic review has yet considered a manipulation check in assessing the effects of single bouts of physical exertion on executive function.

Given the lack of a systematic evaluation of dose-related PA effects on executive functions considering the measurement of fitness gains after long-term interventions as well as the lack of short-term studies considering the analysis of experimental manipulations, the scientific gap should be filled. In this context, the primary objective of the present systematic review was to compare the dose-related effects of PA interventions on executive functions in healthy schoolchildren aged 6–12 years, including data on fitness development after long-term interventions, while considering the success of experimental manipulations in the short-term interventions. Due to the large diversity of PA interventions and effects, this evaluation focused on the impact of short- and long-term endurance, strength, and coordination training on executive functions.

The review questions were:(1)Is there any effect of short-term endurance, strength, or coordination training on the selected executive parameters after controlling for the experimental success?(2)Is there any effect of long-term endurance, strength or coordination training on the selected executive parameters considering changes in fitness performance?(3)Does the amount of training have a modulating influence on the outcome?(4)Does the outcome depend on the success of the experimental manipulation?

## 2. Materials and Methods

This systematic review was preregistered (PROSPERO registration number: CRD42021239242) at the international prospective register of systematic reviews and meta-analysis and followed the Preferred Reporting Items for Systematic Reviews and Meta-Analyses (PRISMA) guidelines [20].

### 2.1. Search Strategy

Article selection was conducted by two independent researchers (AD and KS) from May 2021 to September 2021 in the following electronic databases: MEDLINE via PubMed and Web of Science, Cochrane Library, Scopus, Eric (Boolean searches using operator “OR” and “AND”), as well a complementary citation search in Deutscher Bildungsserver, Research Gate and Google Scholar (citation and research group tracking). Articles published in English and German were considered for inclusion without date restriction. If there was no access to the free full-text articles, the corresponding author was contacted.

For this study, it was necessary to define the term used in the context of PA intervention and executive functions. A detailed description follows below (Table 1 and Table 2). The search strategy (Boolean logic) included various combinations of keywords related to PA (including endurance, strength, and coordination training), executive functions (including inhibitory control, working memory and cognitive flexibility) and schoolchildren aged 6–12 years combined in Boolean logic. The following search term was used: “physical activity” OR “exercises” OR “sport” OR “training” OR “gymnastics” OR “workout” OR “games” OR “coordination training” OR “physical endurance” OR “motor fitness” OR “strength” OR “speed” OR “power” OR “aerobics” OR “balance” OR “running” OR “cardiorespiratory fitness” OR “cardiovascular endurance” OR “muscular endurance” OR “agility” OR “rapidity” OR “resistance exercise” OR “locomotor skills” OR “object control skills” OR “coordinative” OR “aerobic capacity” OR “muscular fitness” AND “executive function” OR “inhibition” OR “inhibiting ability” OR “self-control” OR “working memory” OR “updating” OR “cognitive flexibility” OR “task switching” OR “shifting” AND “child” OR “student” OR “schoolchildren” OR “children” OR “preadolescents”.

The PICO tool, which focuses on population, intervention, comparison, and outcomes, was used to track the selection criteria of RCT. Two reviewers (AD, KS) independently reviewed the results obtained from the electronic database search in a stepwise procedure. First, the titles were checked for admissibility, which was followed by abstract and full text screening. In addition, the reviewers searched for references and citations from included studies, relevant reviews, or meta-analyses (citation and research group tracking).

### 2.2. Selection Criteria

Articles were included if both reviewers reached a consensus on eligibility. The third reviewer (GJ) was invited to discuss if no consensus had been reached (i.e., clarification of discrepancies in data collection, PA measurement). Studies that met the following criteria were considered for inclusion: (1) RCTs without or with a cross-over strategy (within-subject design), (2) studies with any control condition, (3) studies on typically developing schoolchildren between 6 and 12 years of age (upper limit for the growth age according to the Tanner stage 2, range: 8.2–12.1 years [25]; without restrictions on weight, sex, culture, religion or social conditions, (4) studies examined short-term effects of a single bout of PA or long-term/chronic PA (repeated bouts of PA over weeks or months) on executive function, (5) the short- and long-term interventions only include PA related to endurance, strength or coordination, (6) only studies with pre–post measurement of intervention effects, (7) measurements of core executive functions with computerized assessment tools or its modified paper version, (8) long-term studies should include a pre–post measurement of physical fitness, and (9) short-term intervention should verify the success of the experimental manipulations (monitoring of physical exertion).

Studies were excluded if (1) the study population consisted of children with mental or cognition disorders, nervous system diseases, brain injuries and other disorders that could affect cognition, (2) the study population consisted of pre-school and kindergarten children, (3) studies had no control group, (4) the intervention consisted of a complex PA or physically active video games without distinguishing between endurance, strength, and coordination.

#### 2.2.1. Characteristic Features of Intervention

The PA intervention was defined in terms of endurance, strength and coordination and compared with the international terms. In Germany, “motor skills” are equated with “fitness skills” [21,22], but this is not consistent with international terminology [23,24]. Therefore, only the term “fitness skills” was used in this review, which refers to endurance, strength, and coordination. Analogies to the terms are listed in Table 1.

#### 2.2.2. Characteristic Features of Executive Functions

To narrow the scope of the review, the research search was limited to the 3 core executive functions (inhibitory control, working memory and cognitive flexibility) (Table 2) [1].

### 2.3. Data Extraction

After the first selection of articles and extraction of duplicates, two reviewers (AD, KS) independently extracted the main characteristics of the included studies: main author, year of publication, country of performance, population characteristics (sample size, mean age, sex distribution), type of intervention and control group, characteristics of the PA intervention and control session (period, amount, intensity, frequency), assessment and measurement tool, and main findings. A difference was made between the measurements for short- and long-term experimental manipulations. Short-term interventions aimed to capture the success of experimental manipulation (physical and cognitive exertion), while long-term interventions focused on pre–post physical fitness gain as well as the measurement of compliance (attendance rate). Regarding the age of the population, studies with an age range other than 6–12 years, e.g., participants aged 5–18 years, could only be included if a subgroup analysis for school children aged 6–12 years was available separately. If only the average age of the population was reported, the Tanner stage 2 [25] for the prepubertal developmental phase was the prerequisite for inclusion in the analysis. Children aged 13 years and above were excluded.

### 2.4. Risk of Bias/Quality Assessment

The quality of included studies was assessed using the Cochrane Collaboration’s tool for risk of bias in randomized trials [26]. This tool examines selection bias (random sequence generation and allocation concealment), performance bias (blinding participants and personnel), detection bias (blinding of outcome assessment), attrition bias (participants lost during study) and reporting bias (selective outcome reporting of prespecified outcome measures in methods sections) and other bias (trial methods). For each of these bias categories, studies were classified into low, unclear, or high risk of bias. Unclear biases were used for an inadequate description. In addition, selection bias was defined as unclear when the power of the sample size was not calculated. Performance bias was considered high if the study staff was the same for each treatment (intervention group and control group) or if the school teacher supervised the intervention (possible Hawthorne effect) [27].

### 2.5. Strategy for Data Synthesis

In the included studies, a qualitative and quantitative synthesis was used regarding the short- and long-term interventions. These data were synthesized in a table for analysis and compared according to the dose-dependent effects of the intervention. Data analysis was carried out by a single operator (AD) and verified by a second (KS).

## 3. Results

### 3.1. Selected Articles and Characteristics

The electronic data search yielded 1774 articles, of which 17 RCTs met all inclusion criteria (Figure 1). The full text of one article could not be accessed. The characteristics of included studies in terms of short- and long-term interventions are shown in Table 3, Table 4 and Table 5. Nine studies used a single bout of PA as part of the intervention, and eight used repeated bouts of PA over weeks or months. The studies were conducted in nine countries: one in Canada [28], one in the United Kingdom [29], three in Switzerland [30,31,32], five in the United States [33,34,35,36,37], one in China [38], one in Spain [39], two in the Netherlands [40,41], two in Germany [42,43], and one in Italy [44]. The randomly allocated interventions were carried out in school settings and outside of school (laboratories and sports area). Endurance training was mainly used in short-term interventions, while long-term interventions focused on coordination. No short- and long-term strength training interventions were found, so the strength was not addressed further.

### 3.2. Short-Term Interventions

In these studies, sample sizes ranged from 20 to 309 children with an approximately equal distribution of boys and girls (average percentage of girls 49.6%, range 40.0–64.6%). Two studies used “stratified” sex-specific randomization techniques and four studies used a within-subjects design with a washout period of at least one week.

Most short-term PA interventions were based on endurance training (7/9), with a small proportion investigating endurance training versus coordinative sessions (2/9), and lasted between 5 and 30 min. The coordinative training was carried out twice separately (2/9). The control groups received predominantly sedentary treatment with or without cognitive challenges (8/9). In one study, the comparison group received treatment with low physical exertion and low cognitive engagement.

The methods to assess physical exertion during the intervention were cardiorespiratory/aerobic capacity through heart rate monitoring, accelerometer-based PA level, computerized indirect calorimetry (VO_2_max), and subjective scoring of PA intensity. In most studies, only the average intensity values for the groups were reported. One study reported the measured range of heart rate in the children during the intervention (57.7% to 80.1% of age-predicted maximal HR) [34], while two studies implemented personalized intensities of PA for single children [35,36]. Most studies documented successful experimental manipulation (7/9), but only one study evaluated cognitive engagement in this context [30].

Measures of executive function included inhibitory control, working memory and cognitive flexibility, most of which were taken immediately after treatment or 5 min later (5/9). In other studies, the tasks were performed at least 10 min after the intervention (4/9).

### 3.3. Short-Term Effects of the PA Intervention on Executive Functions

Intervention effects on children’s executive functions were observed in six studies [30,31,34,35,36,38], including a study with negative effects of a 20 min intervention with cognitive demands [30] (Table 1). One study found a positive effect of endurance training at moderate intensity (30 min condition, 60–70% of HR max) on the reaction speed of all core executive functions [38]. The other four studies only showed positive effects on inhibitory control (10–20 min condition, moderate to vigorous activity, at least 60% of HR max) after endurance training [35,36] and coordinative training [31,34]. One study investigated possible differential effects on executive functions when controlling for experimental manipulation parameters [31]. For inhibition, the study results showed no changes after controlling for heart rate as a measure of intervention intensity.

### 3.4. Long-Term Interventions

These studies included sample sizes from 45 to 510 children with an approximately equal distribution of boys and girls (average percentage of girls 49.8%, range 44.4–54.9%). In four cluster trials, whole school classes were randomly assigned to the intervention and control groups.

Most long-term PA interventions were based on coordinative training (6/8) and lasted between 6 and 9 months. During this period, PA interventions of 10 to 70 min were conducted either every school day (five times a week) or at least twice a week. The endurance training was conducted separately (2/8) or in comparison to the coordinative training (3/8) and lasted between 6 and 44 weeks. Each intervention (at least twice, at most five times a week) consisted of a 10 to 45 min activity session. The control groups either received no treatment (4/8) or were randomly assigned to an active control treatment (4/8). In one study, a wait list control group received the intervention later [37].

Pre–post measurements of endurance (aerobic fitness or cardiorespiratory endurance) were performed using tests on a treadmill, shuttle run test (stages), and VO2max (computerized indirect calorimetry or estimated); coordination by coordinative performance (dribbling, Heidelberg Gross Motor Test), speed coordination (10 × 5 m shuttle run), speed by 25 m sprint, and strength through standing long jump, handgrip and sit-ups. While aerobic fitness was tested in all studies, coordinative skills were tested in three studies (Table 5). Three studies lacked a follow-up measurement of coordination ability [32,37,44], and three others reported a lack of fitness gains [40,41,42] within the respective intervention groups.

Measures of executive functions included inhibition, working memory, and cognitive flexibility.

### 3.5. Long-Term Effects of the PA Intervention on Executive Functions

All studies showed an increase in executive function performance from baseline to post-intervention, indicating period effects. The lack of intervention effects on executive functions was associated with the lack of specific fitness gains [40,41,42,44]. In comparison, two 10-week coordination interventions showed an advantage in motor skills and EF compared to the control or endurance group [39,43]. Shorter 6-week PA intervention with a successful improvement in aerobic fitness documented an increase in shift performance but only in the coordinative group [32]. Improved endurance performance after 9 months of coordinative training showed a causal relationship with inhibition and cognitive flexibility [37].

### 3.6. Risk of Bias within Studies

The methodological quality of the included studies is summarized in Table 6. Most studies described the method used to generate the randomized allocation sequence. However, seven out of 17 studies did not describe allocation concealment. Unclear selection bias was found in eight studies because power calculations for appropriate sample sizes were not provided. Most studies (15/17) did not blind participants and/or assessors. In fact, performance bias has been identified as the most frequent bias due to the practical difficulties of testing children in school or PA interventions. There were seven studies that did not describe the blinding of outcome assessment and were therefore classified as unclear. Mostly, a low risk of bias in results and reporting was found. Other unclear biases resulted from possible learning effects or loss of motivation in studies with within-subjects design. In one study, validated measurements of PA intensity were obtained in only 7.2% of participants [38], and in another one, only the subjective measurement of PA intensity was used [33].

## 4. Discussion

This study is the first systematic review to address the dose-related effects of short- and long-term endurance, strength, and coordination training on executive functions in healthy children aged 6–12 years, considering the success of experimental manipulations and fitness development. Given that few studies demonstrated no effects, the results suggest that short-term endurance and coordination training can improve inhibitory control after a successful experimental manipulation, but not working memory and cognitive flexibility, while long-term PA intervention seems to have a positive effect on three core executive functions consistent with improved fitness performance. No studies on strength training in children were found that met the inclusion criteria. In general, the intervention studies with children varied in their methodology, as well as in the different risks of bias, which have been discussed several times in the past [3,7,8,46].

### 4.1. Effects of Short-Term PA Training

The novelty of the current review compared to the previous evaluations was the analysis of experimental manipulations in the included studies. Without the successful implementation of the PA interventions, i.e., verification of physical exertion with appropriate methods, it is unlikely that effects will occur [46]. Considering various methods of measuring physical exertion, most included studies demonstrated successful experimental manipulation. Although the implementation of PA intensity was unclear in two studies [33,38], this did not alter the conclusions of this evaluation. In line with others [7,8,45,47], current results on the causal relation between specific physical exertion and executive functions are inconsistent. This is particularly noticeable in interventions that last 20 min. Regardless of whether it was endurance or coordination activity, a 20 min training session did not seem to have a positive impact on working memory and cognitive flexibility. One study even suggested that 20 min of activity with high cognitive demands overwhelms children under the age of 9 in the shifting task [30]. The authors therefore point to age-appropriate physical exertion to achieve positive effects [28,30]. Because RCTs with shorter interventions than 20 min analyzing the effects on working memory and cognitive flexibility in children up to 9 years are missing, no clear conclusions can be drawn about activity duration for this age group. In contrast to working memory and cognitive flexibility, studies have shown that 20 min of PA (both endurance and coordination) has a positive effect on inhibitory control in children aged 9 years and older [31,34,35,36,38] and even younger [31]. Similar results have been reported previously [45] and could therefore be relevant to school challenges. In a previous meta-analysis, the authors indicated that age may play a moderating role in the inhibition benefits induced by a single bout of aerobic exercise [48]. It has been suggested that younger children are more sensitive to external stimuli such as PA than adolescents due to their immature executive system. However, the small number of studies with pre-adolescent children and an inclusion of only moderate-intensity aerobic sessions in the aforementioned study limited the significance, which is similar to the present analysis.

A comparison of only three available studies with shorter sessions than 20 min showed that a coordinative session of 10 min improved inhibitory control, but not an endurance session of 5, 10 or 15 min [29,33,34]. However, the range of training intensity in studies (moderate to vigorous) and the measurement methods used varied. Two studies with similar study designs have shown that 60% of the maximum heart rate for 20 min on the treadmill is sufficient to improve inhibitory control [35,36]. Once the intensity range varied from moderate to vigorous activity, intervention effects were often no longer observed. An exception was the only study with an intervention period of 30 min and a beneficial effect on the reaction speed of all three executive functions but not on accuracy [38]. Because this study had several risks of bias, particularly the lack of measurement of physical activity intensity in all participating children, these results should be viewed with caution. In the only study controlling for the modulating effect of PA intensity on inhibition, heart rate had no effect on the results [31]. However, the authors showed that cortisol increase correlated with inhibitory performance independent of heart rate. Interestingly, the physical exertion of the children affected cortisol levels differently. Responders and non-responders were mentioned as explanations for the results. In another study with children aged 9–10 years, no exercise-induced changes in cortisol were found, suggesting specific psychological stressors at this age [49]. Both studies thus revealed new perspectives for the evaluation of experimental manipulations.

### 4.2. Effects of Long-Term PA Training

This evaluation addressed new scientific questions about the long-term effects of the PA intervention, considering changes in fitness performance. It has been observed that a PA intervention of several weeks seems to have a positive impact on executive functions when fitness skills are improved. The benefits that occurred could be related to the experimental manipulations of endurance and coordinative training. No clear indications for the duration and frequency of PA could be found, as the benefits of exercise on executive functions were documented after different periods. Thus, the assumption “the more PA the better” could not be verified. For example, the longest intervention study showed no cognitive benefits over two years [41]. More crucial in this context seems to be the successful experimental manipulation and the expected fitness gains. In the few studies that did examine fitness gains at all, there were studies that lacked tests to verify coordinative progress [32,37,44]. It was unclear whether the EF benefits were due to the coordinative improvements because only the VO2max was tested. The results thus show the importance of the manipulation check and the subsequent verification of fitness after the intervention. Another study with successful manipulative monitoring for coordinative sessions only showed an improvement in inhibition in overweight children in the same intervention group [44]. Although there are similar findings in overweight children [13], it was surprising that after 21 weeks of intervention with an average heart rate of 150 bpm, there was no improvement in aerobic fitness in the whole group. This was contrary to the results of others, who were able to demonstrate an improvement in both VO2max and shifting performance after only 6 weeks of intervention with similar training intensity [32] or an improvement in shuttle running and working memory after 10 weeks [43]. However, the average improvement in VO_2_max of less than 4% did not seem sufficiently effective for cognition, while 4.7% and 5.6% did [32,37]. Despite missing data on coordinative gains, Schmidt and colleagues [32] showed significant differences in the use of executive functions during the training sessions across intervention and control groups. Team games in the coordinative group led to significantly higher use of attention, memory and cognitive flexibility compared to the other groups. The benefits of team games have been shown in three other studies as well, two after 30 sessions within 10 weeks each and one after 9 months of daily games [37,39,43]. Nevertheless, the insufficient or lack of improvement in fitness skills across the different intervention groups made evaluation difficult. Consequently, the coordinative and the endurance sessions could not be adequately compared.

Further attention was paid to the attendance rate of the intervention in relation to the dose-dependent effects suggested elsewhere [46]. In general, the data showed that the average attendance was high. However, studies showed that participation in the intervention varied between 49% and 98% [40], and sometimes less than 40% [37], which may lead to different results than expected. Hillman and colleagues used the attendance rate as a covariate to examine the dose-related association with cognitive development [37]. They demonstrated a dose–response relation between participation in the PA program and executive control with simultaneous positive electrical changes in brain activity. Given that the 9-week intervention in the other study had no effect on cardiovascular fitness or EF [40], attendance rate in relation to results would be of interest, especially since the manipulation check of the same study showed that the scheduled 10-min moderate–vigorous PA lasted only 2.9 min. It can be suggested that the 9-week intervention failed to meet the requirements for experimental implementation. To better understand the missing effects of PA programs, it is therefore important to pay more attention to manipulation check and attendance rates.

As mentioned above, interventions do not seem to have been successful only by increasing training intensity or duration, as other authors have suggested [41,42,44]. Instead, specific workouts and sustained challenging exercises might be more helpful in promoting executive functions. A previous evaluation of children aged 6 to 12 years found effect differences between long-term PA programs in favor of coordinative and cognitive exertion [45]. Others found similar results [7,47]. However, no other moderators of dose-related effects were considered in either evaluation. Furthermore, a mediator analysis considering the fitness skills of the participants before the intervention only partially clarifies the results [32,44]. Pedro, et al., pointed to the controversial relation between improvement in aerobic capacity (VO_2_max) and executive functions [39]. Koutsandreous, et al., showed that heart rate during training did not affect outcomes, but varied and challenging team games did [43]. Thus, the consideration of other modulators in the assessment (e.g., fitness gain and heart rate) seems valuable and complementary.

The current analysis was challenging due to the small number of studies to answer the research questions. Based on only nine and eight studies of short- and long-term interventions, respectively, long-term PA appears to be more effective than short-term PA in improving executive functions. Higher-order cognitive abilities, such as working memory [50], may take longer to undergo physiological changes. Regardless of the type of intervention, all long-term studies in this analysis showed an effect of time on working memory in contrast to short-term interventions. However, further RCTs are needed to determine the dose-related effects of short- and long-term PA interventions on executive functions.

### 4.3. Perspective

There is not yet sufficient evidence on the relation between different types of sport and executive functions in school-aged children to formulate dose- and type-specific guidelines for physical activity. This review highlights the importance of high-quality intervention studies that use various measurement tools to assess physical exertion in children and track fitness development. It is noted that most intervention studies with school children do not include a dose-related evaluation of physical activity or a quality assessment of the physical activity implementation, resulting in insufficient evidence for the success of the intervention. Given the high variability in children’s fitness and cognitive performance, physical activity in RCTs should be more focused on individual performance. Overwhelming or unchallenging activities may result in insufficient benefits for children.

## 5. Conclusions

Our systematic review of RCTs on dose-related effects of PA and executive functions remains limited due to the small number of studies. The quality of future studies could benefit from the assessment of the experimental manipulation and the verification of the specific fitness results. Overall, there seems to be a causal relation between short- and long-term PA programs and executive functions, especially inhibitory control. Regardless of the short-term PA type; a single 20-min training session can either overwhelm or enhance short-term executive functions, depending on the age of the schoolchildren. Long-term PA interventions should focus on challenging PA programs with fitness gains.

## Figures and Tables

**Figure 1 children-09-01651-f001:**
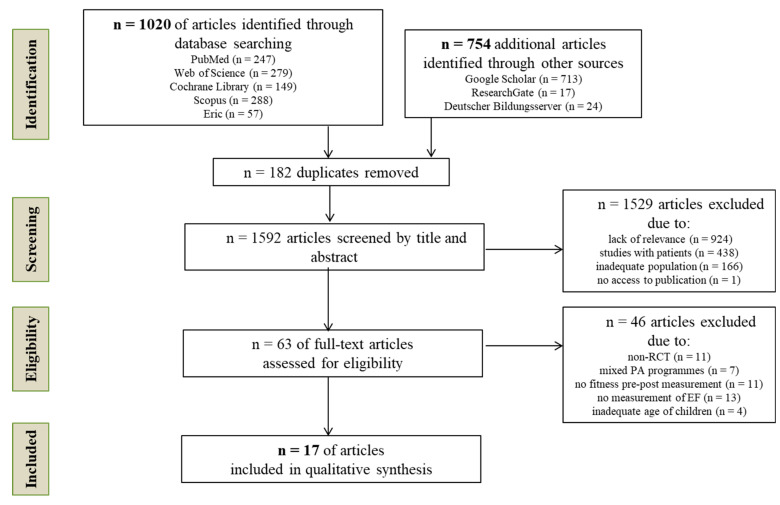
PRISMA flow diagram of studies through the review process.

**Table 1 children-09-01651-t001:** Definitions used for the domains of skill-related physical fitness (adapted from literature) [21,22,23,24].

Intervention of Physical Activity (Training, Gymnastics, Exercise, Workout)	
Endurance	Endurance is the ability to withstand a load physically and mentally over a longer period, which due to its intensity and duration leads to insurmountable fatigue, and to recover as quickly as possible/non-cognitively engaging physical exertion (cardiorespiratory fitness, cardiovascular endurance, muscular endurance, aerobic fitness)
Strength	Strength describes the ability of muscles to resist, counteract, or hold resistance, and quick strength describes the ability to perform movements as quickly as possible against a resistance (muscular strength, resistance exercise, power, agility, speed, rapidity, vigorous activity)
Coordination	Coordination is the ability to perform challenging movements quickly and purposefully with high quality/cognitively demanding exercises (motor fitness, balance, control of body movement, psychomotor ability, team games)

**Table 2 children-09-01651-t002:** Definitions used for the domains of executive functions (adapted from literature) [1].

Executive Functions (Higher-Order Cognitive Abilities)	
Inhibitory control	Inhibitory control means resisting the initial impulse or strong need to do something and instead thinking first and then acting (self-control and interference control incl. selective attention); Tasks: Stroop task, Simon task, Flanker task, antisaccade tasks, go/no-go task, and stop-signal task
Working memory	Working memory (WM) refers to the ability to hold new information in memory while performing mental operations related to the information held (verbal WM and visual-spatial WM); Tasks: Hearts and Flowers task (Dots task), N-back task, Corsi Block task, Backward-digit span or mixed-digit order
Cognitive flexibility	Cognitive flexibility is the ability to flexibly adjust attention to changing demands or priorities, or to change the way one looks at things (shifting, mental flexibility, or mental set shifting and closely linked to creativity); Tasks: task-switching and set-shifting tasks, the Trail-Making Task

**Table 3 children-09-01651-t003:** Short-term physical activity intervention.

Study (Reference, Year, Country)	Sample (*n*, Age, % Girls)	Setting/Design	Intervention (Conditions)	Experimental Manipulation (Assessment)	Cognition (Assessment, PostTest Time)	Findings
Bedard et al., 2021, Canada, [28]	*n* = 48, 6–8 years, 40.0% girls	Lab/RCT	**Endurance**: non-cognitively engaging PA (running); 20 min conditions **Coordination**: cognitively engaging PA (running to the board to play the game, approximately 75% of HR max; 135–160 bpm); 20 min conditions **Control**: cognitive sedentary activity (board game); 20 min conditions	**EM**: successful -HR -Feeling scale -Borg RPE scale	Inhibition (Flanker task) **Time**: 10–15 min of the end of treatment	No intervention effects
Chen et al., 2014, China, [38]	*n* = 83, 9 and 11 years, 50.6% girls	School/ RCT stratified by sex and grade	**Endurance**: jogging at moderate intensity 60–70% of HR max, ≈157 bpm; 30 min condition **Control**: 30 min of sitting quietly and reading book	**EM**: unclear -HR (based on 6 children per group)	Inhibition (Flanker task), working memory (2-back task), and shifting (a more-odd task) **Time**: 20–25 min of the end of treatment	For all tasks: shorter response time in the endurance group. No intervention effects on accuracy
Drollette et al., 2012, USA, [35]	*n* = 36, 9–11 years, 55.6% girls	Lab/within-subject repeated measures (each participant as their own control: 8.4 days a washout period)	**Endurance**: treadmill walking (3 testing periods: before, during walking and post), moderately intense (60% of max HR); ≈20 min condition, **Control**: seated rest on the treadmilll (3 testing periods: before, during seated rest and post seated); ≈20 min condition	**EM**: successful -HR -indirect calorimetry (VO_2_max) -RPE	Inhibitory control (Flanker task), working memory (spatial n-back task) **Time**: about 5 min after walking or seating	Post-exercise-increased inhibitory control. No intervention effects on working memory
Egger et al., 2018, Switzerland, [30]	*n* = 216, 7 and 9 years, 49.1% girls	School/RCT	**Coordination**: (a) Combo group (high CE, high PE, 67% of HRmax), (b) Cognition group (high CE, low PE, 47% of max HR); 20 min condition **Endurance**: (c) Aerobic group (low CE, high PE); High PE: 67% of HR max; 20 min condition **Control**: low CE, low PE (47% of HRmax); 20 min condition	**EM**: successful -HR -Perceived cognitive engagement -Borg RPE scale	Inhibition (Flanker task), updating (a Backward Color Recall task) and Shifting (an additional “mixed” block within the Flanker task) **Time**: immediately after the treatment	No intervention effects for PE factor. Factor CE affected negative shifting
Hillman et al., 2009, USA, [36]	*n* = 20, mean age 9.6 years, 40.0% girls	Lab/within-subject repeated measures (each participant as their own control: 10.6 days a washout period)	**Endurance**: treadmill walking, moderately intense (60% of max HR, ~125.4 bpm); 20 min condition **Control**: 20 min of the resting session	**EM**: successful -HR -indirect calorimetry (VO_2_max) -RPE	Inhibitory control (Flanker task **Time**: 25 min post exercise	Post exercise increased response accuracy
Howie et al., 2015, USA, [33]	*n* = 96, 9–12 years, 64.6% girls	School/within-subject design (Latin Square design), washout period at least 1 week	**Endurance**: classroom exercise break (jumping and running in place, moderate-to-vigorous PA); 5 min, 10 min and 20 min conditions, **Control**: 10 min of sedentary classroom lesson about exercise science	**EM**: unclear -Observing fitness instruction time based on videotapes	Executive control (modified Trail-Making-Test), working memory (Digit Recall in chronological order, paper based) **Time**: immediately thereafter	No intervention effects.
Jäger et al., 2014, Switzerland, [31]	*n* = 104, 6–8 years, 54.8% girls	School/RCT	**Coordination**: cognitively engaging and playful physical activity (three games: (1) running with different movements to a song, (2) playing tag to different rules and (3) rope, club, ball, rod, hula-hoop), moderate-to vigorous activity: HR ≈157 bpm; 20 min condition **Control**: 20 min resting condition listened to a story	**EM**: successful -HR -Enjoyment scale -Cortisol	Updating (n-back task), inhibition and shifting (Flanker task) **Time**: immediately after (post-test) and 40 min after (follow-up)	Positive effects on inhibition, but not on updating and shifting
Morris et al., 2019, UK, [29]	*n* = 303, 9–11 years, 37.3% girls	School/ 4-block randomization stratified by sex	**Endurance**: the Daily Mile: TDM in the playground program (an additional 15 min of walking or running, at least 10 min of moderate-to-vigorous intensity) **Control**: usual classroom-based academic lesson	**EM**: successful accelerometer-based PA level: -sedentary -light -moderate-to-vigorous	Inhibitory control (modified Flanker and Animal Stroop task: paper-based) working memory (Digit Recall in chronological order) cognitive flexibility (Trail-Making Task) **Time**: within 5 min of the end of treatment	No intervention effects
Vazou et al., 2014, USA, [34]	*n* = 35, 9–11 years (54.3% girls)	LAB/a within-subjects design with 7.8 days between sessions	**Coordination**: movement while avoiding obstacles with math practice (sliding, hopping, leaping, bear or crab walking etc.), 68.41% of age-predicted maximal HR: ~143 bpm; 10-min condition at least moderate intensity **Control**: a seated math practice, cognitively challenging	**EM**: successful -HR -PA enjoyment scale -RPE	Inhibition, working memory, switching, and selective attention (the Standard Flanker, Reverse Flanker, and Mixed Flanker), **Time**: immediately thereafter	Post exercise response time in the Standard Flanker improved

Bpm beats per minute, CE cognitive engagement, EM experimental manipulation, HR heart rate, PA physical activity, PE physical education, RPE rating of perceived exertion.

**Table 4 children-09-01651-t004:** Long-term physical activity intervention.

Study (Reference, Year, Country)	Sample (*n*, Age, % Girls)	Setting/Design	Intervention (Conditions)	Fitness Gain (FG) (Assessment)	Cognition (Assessment)	Findings
Crova et al., 2014, Italy, [44]	*n* = 70, 9–10 years, 50.0% girls	School/ a class-based cluster-RCT	**Coordination**: two additional PE hours of skill-based and tennis-specific training (HR > 139 bpm); **6 month** (21 weeks, 2 h a week) **Control**: regular PE	**FG**: unclear (coordination not tested) Aerobic fitness (20 m Shuttle Run test, estimated VO_2_max)	Inhibition and working memory updating (random number generation task)	Effect of time on inhibition; effect of intervention Covariate: BMI and VO_2_max at pretest related to inhibition
De Greeff et al., 2016, The Netherlands, [41]	*n* = 499, mean age 8.1 years, 54.7% girls	School/ a class-based cluster-RCT stratified by grade	**Endurance**: MVPA during academic lessons (jogging, hopping, marching), ≈60% of HRmax; 2 school years; **22 weeks program per year**, 3 lessons per week, 20–30 min per lesson **Control**: regular lessons	**FG**: successful for speed coordination (coordination not tested). Speed coordination (10 × 5-m Shuttle Run) aerobic fitness (20 m Shuttle Run), and muscular fitness (standing long jump, sit-ups, handgrip strength)	Inhibition (the Golden Stroop test), working memory (the Digit and Visual span backward), and cognitive flexibility (the Wisconsin card-sorting test)	Effect of time; no intervention effects Covariate: none considered
Hillman et al., 2014, USA, [37]	*n* = 221, 7–9 years, 46.2% girls	After school program/RCT	**Coordination**: aerobically demanding PA and low organizational games to refine motor skills (HR ≈137 bpm); **9-month** (150 days of the school year, each school day, 70 min) **Control**: a wait-list group	**FG**: unclear (coordination not tested) aerobic fitness (computerized indirect Calorimetry, VO_2_max)	Inhibition (modified Flanker task) and cognitive flexibility (color–shape switch task)	Effect of time; effect of intervention Covariate: PA attendance related to specific EF
Koutsandreou et al., 2016, Germany, [43]	*n* = 71, 9–10 years, 54.9% girls	After school/ RCT	**Coordination**: motor-demanding exercise (team games); HR ≈125 bpm **Endurance**: cardiovascular exercise (running without any high motor demand, HR ≈139 bpm); **10 weeks**, 3 times/week for 45 min **Control**: assisted homework session	**FG**: successful Motor fitness (Heidelberg Gross Motor Test) and cardiorespiratory endurance (20 m Shuttle Run Test)	Working memory (Letter Digit Span with mixed-digit order)	Effect of time; effect of intervention Covariate: HR not related
Ludyga et al., 2019, Germany, [42]	*n* = 45, 9–10 years, 44.4% girls	After school/ RCT	**Endurance:** aerobic training with running-based games (HR ≈140 bpm) **Coordination:** fine and gross motor body training (HR ≈124 bpm); **10 weeks**, 3 times/week for 45 min **Control**: assisted homework sessions to prevent attention bias	**FG**: only successful for aerobic fitness. Motor fitness (total score of the Heidelberg Gross Motor Test) and aerobic fitness (stages on the 20 m Shuttle Run)	Inhibitory control (Flanker task)	Effect of time; no intervention effects Covariate: none considered
Pedro Ángel et al., 2021, Spain, [39]	*n* = 114, 8–12 years, 47.3% girls	School/RCT	**Coordination**: PA with cognitive engagement (team games, moderate to vigorous intensity and recovery periods: Borg scale 6–10); **10 weeks**, 3 times a week for each 30 min) **Control**: regular lessons	**FG**: successful for dribbling and aerobic fitness (20 m Shuttle Run test, number of stages and estimated VO_2_max), lower body strength (standing long jump), sprint, handgrip strength, motor skills (dribbling performance)	Cognitive flexibility (Trail-Making Test, paper-based)	Effect of time; effect of intervention Covariate: ∆VO_2_max related positive
Schmidt et al., 2015, Switzerland, [32]	*n* = 181, 10–12 years, 54.7% girls	School/ a class-based cluster-RCT	**Coordination**: high cognitive engagement (team games) **Endurance**: aerobic exercise (low cognitive engagement); **6 weeks**, 2 PE per week, 45 min each **Control**: low physical exertion and low cognitive engagement	**FG**: successful for VO_2_max aerobic fitness (20 m Shuttle Run test, stages and estimated VO_2_max)	Updating (n-back task), inhibition (Flanker task), shifting (“mixed” block included in the Flanker task)	Effect of time; effect of intervention Covariate: baseline VO_2_max not related
van den Berg et al., 2019, The Netherlands, [40]	*n* = 512, 9–12 years, 46.5% girls	School/a class-based cluster RCT stratified by grade	**Endurance**: daily exercise breaks with dance movement during classroom time (moderate intensity, 60% of HRmax); **9 weeks** of 10 min breaks per school day (45 exercise breaks) **Control**: 9 educational lessons, lasting 10–15 min, one for each week	**FG**: failed aerobic fitness (modified 18 m Shuttle Run test and estimated VO_2_max)	Inhibition (Stroop Color–Word task) and interference control (Attention Network Task)	Effect of time; no intervention effect Covariate: none considered

Bpm beats per minute, HR heart rate, MVPA moderate-vigorous physical activity, PA physical activity, PE physical education.

**Table 5 children-09-01651-t005:** Long-term physical activity intervention.

Studies	Period	Frequency	Duration	PA Sessions	Experimental Manipulation	Attendance (A), Manipulation Check (MC)	Fitness Gain Differences	Executive Gain Differences
Crova, et al., 2014, [44]	21 weeks	2 times/wk	60 min	Coordination	Learning novel skills	**A**: na, **MC**: HR 150 bpm, specialized skills ↑	VO_2_max Ø	Memory Ø, Inhibition ↑ only in overweight
de Greeff, et al., 2016, [45]	44 weeks	3 times/wk	20–30 min	Endurance	MVPA	**A**: 88.6%, **MC**: 14 min of MVPA	Score SH Ø, Speed ↑	3 core EF Ø
Hillman, et al., 2014, [37]	9 months	5 times/wk	70 min	Coordination	MVPA, refine motor skills	**A**: 80.6%, **MC**: HR 137 bpm, ≈4246 steps	5.6%↑ of VO_2_max	Inhibition↑ Cognitive flexibility ↑
Koutsandreou, et al., 2016, [43]	10 weeks	3 times/wk	45 min	Coordination Endurance	Constantly challenging MVPA	**A**: ≈94%, **MC**: HR 125 bpm A: ≈94%, MC: HR 139 bpm	MF↑ Score SH↑	Working memory ↑↑ Working memory ↑
Ludyga, et al., 2019, [42]	10 weeks	3 times/wk	45 min	Coordination Endurance	Unclear complexity MVPA	**A**: ≈27 sessions, **MC**:HR 124 bpm A: ≈28 sessions, MC:HR 140 bpm	MF Ø Score SH↑	Inhibition Ø Inhibition Ø
Pedro, et al., 2021, [39]	10 weeks	3 times/wk	30 min	Coordination	Borg scale 6–10 points	**A**: ≈96%, **MC**: Borg scale ≈6.9 points	Dribbling↑	Cognitive flexibility ↑
Schmidt, et al., 2015, [32]	6 weeks	2 times/wk	45 min	Coordination Endurance	Mental control, MVPA MVPA	**A**: ≈11 lessons, **MC**: HR 148 bpm, mental rate ↑ A: ≈12 lessons, MC: HR 150 bpm, mental rate ↓	4.69%↑ of VO_2_max 3.79%↑ of VO_2_max	Shifting ↑ 3 core EF Ø
van der Berg, et al., 2019, [40]	9 weeks	5 times/wk	10 min	Endurance	MVPA	**A**: ≈89%, **MC**: 2.9 min MVPA	VO_2_max Ø	Inhibition Ø

MVPA moderate–vigorous physical activity; HR heart rate, MF motor fitness, SH shuttle run Ø no differences in increase; ↑ increasing differences, ↑↑ marked growing differences, *na* not available.

**Table 6 children-09-01651-t006:** Quality assessment.

Cochrane Collaborations	Random Sequence Generation	Allocation Concealment	Performance Bias/Blinding of Participants	Blinding of Outcome Assessment	Attrition Bias Incomplete Outcome Data	Reporting Bias Selective Reporting	Other Bias
**Short-term**							
Bedard, et al., 2021, [28]	Low risk	Low risk	Unclear risk	Unclear risk	Low risk	Low risk	Low risk
Chen, et al., 2014, [38]	Unclear risk	Low risk	High risk	High risk	Unclear risk	Low risk	High risk
Drolette, et al., 2012, [35]	Unclear risk	Low risk	Unclear risk	Unclear risk	Low risk	Low risk	Unclear risk
Egger, et al., 2018, [30]	Low risk	Unclear risk	Unclear risk	Low risk	Low risk	Low risk	Low risk
Hillman, et al., 2009, [36]	High risk	Unclear risk	Unclear risk	Unclear risk	Low risk	Low risk	Unclear risk
Howie, et al., 2015, [33]	Unclear risk	Unclear risk	Unclear risk	Unclear risk	Low risk	Low risk	Unclear risk
Jäger, et al., 2014, [31]	High risk	Unclear risk	Unclear risk	Low risk	Low risk	Low risk	Low risk
Morris, et al., 2019, [29]	Low risk	Low risk	High risk	High risk	Low risk	Low risk	Unclear risk
Vazou, et al., 2014, [34]	High risk	Unclear risk	Unclear risk	Unclear risk	Low risk	Low risk	Unclear risk
**Long-term**							
Crova, et al., 2014, [44]	Uclear risk	Low risk	Unclear risk	Unclear risk	High risk	Low risk	Low risk
de Greeff, et al., 2016, [45]	Low risk	Low risk	High risk	Low risk	Low risk	Low risk	Low risk
Hillman, et al., 2014, [37]	Low risk	Low risk	Unclear risk	Unclear risk	Low risk	Low risk	Low risk
Koutsandreou, et al., 2016, [43]	Unclear risk	Unclear risk	High risk	Low risk	High risk	Low risk	Low risk
Ludyga, et al., 2019, [42]	Low risk	Unclear risk	High risk	Low risk	High risk	Low risk	Low risk
Pedro, et al., 2021, [39]	Low risk	Low risk	Unclear risk	Unclear riskk	Low risk	Low risk	Low risk
Schmidt, et al., 2015, [32]	Unclear risk	Unclear risk	Low risk	Low risk	Low risk	Low risk	Low risk
van der Berg, et al., 2019, [46]	Low risk	Low risk	Low risk	Low risk	Low risk	Low risk	Low risk

## Data Availability

Not applicable.

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
