# Peer review of "Dose-Related Effects of Endurance, Strength and Coordination Training on Executive Functions in School-Aged Children: A Systematic Review"

_children, 2022, doi:10.3390/children9111651_

Round 1

Reviewer 1 Report

Thank you for the opportunity to review this article. The aim of the authors was to provide a systematic review on the dose-related effects of physical activities on executive functions in school-aged children. The article is clearly structured, although differences between the various interventions and between the subjects of the various studies make comparison complex, however, the authors have given a detailed description that allows for a fairly detailed overview of the topic. The "Discussion" and "Conclusion" chapters are also clearly written, and it is understood what the authors' intended message is.

I consider this to be a well-presented article, however I have some questions/suggestion for the authors:

- Regarding the Risk of bias, why didn’t they use the Version 2 of the Cochrane risk-of-bias tool for randomized trials (RoB 2), for the quality assessment of the articles?

- The authors distinguished the effects of long and short interventions very clearly, however, I could not find a specification in the text, distinguishing a long-term intervention from a short-term one. I invite them to insert it, or make it more easily findable

- Authors should be more consistent in Tab. 3, particularly regarding the "Study" column (use separation with "/" or ",") or the "Sample" column (age of subjects is expressed differently). I would also suggest to the authors to consider summarizing the information in the "Interventions" column to make it more immediately readable

- In Tab. 5, I encourage the authors to use the same alignment for all cells

- At line 400, please replace "und" with "and"

My appreciation to the authors for their work, I hope they can easily answer my questions.

Author Response

Response to Reviewer 1 Comments

We thank the reviewer for the thorough evaluation of our paper and for the constructive critique and suggestions. We considered all issues addressed and revised our paper accordingly.

General comments

Thank you for the opportunity to review this article. The aim of the authors was to provide a systematic review on the dose-related effects of physical activities on executive functions in school-aged children. The article is clearly structured, although differences between the various interventions and between the subjects of the various studies make comparison complex, however, the authors have given a detailed description that allows for a fairly detailed overview of the topic. The "Discussion" and "Conclusion" chapters are also clearly written, and it is understood what the authors' intended message is. I consider this to be a well-presented article, however I have some questions/suggestion for the authors:

Point 1

Regarding the Risk of bias, why didn’t they use the Version 2 of the Cochrane risk-of-bias tool for randomized trials (RoB 2), for the quality assessment of the articles?

Your question is absolutely justified. Our decision was based on the fact that the 2011 Cochrane Collaboration’s tool (RoB1) simplifies the implementation of such a complex review process by not requiring a comprehensive assessment of bias. The work focused on testing experimental manipulation rather than the risk of bias, which has been described several times in other reviews. We briefly addressed this point in the discussion (Line 334-336). RoB1's tool also provides authors more flexibility in judgment options; example " Basis of judgement" in RoB1 is " Author defined ".

Point 2

The authors distinguished the effects of long and short interventions very clearly, however, I could not find a specification in the text, distinguishing a long-term intervention from a short-term one. I invite them to insert it, or make it more easily findable.

Thank you for this important note. We have added the comparison between a long-term and a short-term intervention in the Discussion section (line 447-454). We have also added the references accordingly (Tiego et al.2018).

Tiego, J.; Testa, R.; Bellgrove, M.A.; Pantelis, C.; Whittle, S. A Hierarchical Model of Inhibitory Control. Front. Psychol. 2018, 9, 1339, doi:10.3389/fpsyg.2018.01339

Point 3

Authors should be more consistent in Tab. 3, particularly regarding the "Study" column (use separation with "/" or ",") or the "Sample" column (age of subjects is expressed differently).

We have revised Tab. 3 and 4. However, age in years is reported as average, range, or specific age, depending on what was reported in each study.

Point 4

I would also suggest to the authors to consider summarizing the information in the "Interventions" column to make it more immediately readable.

The summary of intervention characteristics has now been included as a separate paragraph in the text (Section 3.2, Line 225-230 and Section 3.4, Line 260-267).

Point 5

 In Tab. 5, I encourage the authors to use the same alignment for all cells.

Table 5 has been adjusted and cell contents were centred.

Point 6

At line 400, please replace "und" with "and"

The error has been corrected.

My appreciation to the authors for their work, I hope they can easily answer my questions.

Reviewer 2 Report

Dear authors,

I find the topic chosen for the review very interesting, I congratulate you on your work, but I consider that there are some methodological errors that need to be corrected:

- Google Schoolar and ResearchGate are not reliable databases for a systematic review. 
- Systematic reviews have to be reproducible, with the confusion and number of terms that are exposed as keywords it is difficult to understand the search process. Ideally, a search phrase should be selected and used in all databases to make the process systematic.
- The selection of 63 articles out of 1592 is not clear, it is not specified how many were eliminated with each criterion.

The presentation of results and the discussion are very interesting, but several points of the search strategy and the elimination of unreliable databases used need to be clarified.

I hope these comments will help you to improve your work.

Kind regards.

Author Response

Response to Reviewer 2 Comments

We thank the reviewer for her/his thorough and careful evaluation of our paper and for the constructive critique, comments and suggestions. We considered all issues addressed and revised our paper accordingly.

General comments

Dear authors, I find the topic chosen for the review very interesting, I congratulate you on your work, but I consider that there are some methodological errors that need to be corrected.

Point 1

Google Schoolar and ResearchGate are not reliable databases for a systematic review.

We agree that Google Scholar and ResearchGate are not suitable for systematic search strategy. However, based on recommended search practices (Cooper et al. 2018), we conducted an additional search in these referenced databases (Google Scholar and ResearchGate) by tracking citations and research groups to identify papers not captured in our search strategy. Following the systematic search of keywords (MEDLINE via PubMed and Web of Science, Cochrane Library, Scopus, Eric), citations were searched by analysing the bibliography of each study (backwards citation chasing) and via Google Scholar or ResearchGate (forward citation chasing). Following your suggestion, we corrected the systematic search strategy accordingly (Figure 1). The Google Scholar, ResearchGate and Deutscher Bildungsserver are now defined as other sources (Figure 1).

Cooper C, Booth A, Varley-Campbell J, Britten N, Garside R. Defining the process to literature searching in systematic reviews: a literature review of guidance and supporting studies. BMC Med Res Methodol. 2018 Aug 14;18(1):85. doi: 10.1186/s12874-018-0545-3.

Point 2

Systematic reviews have to be reproducible, with the confusion and number of terms that are exposed as keywords it is difficult to understand the search process. Ideally, a search phrase should be selected and used in all databases to make the process systematic.

You are right. For better understanding and to follow our search strategy (using the Boolean operator “OR” as well as “AND”), we have included in the text that the search terms refer only to the systematic review (via PubMed and Web of Science, Cochrane Library, Scopus, Eric) but not via Google Scholar, ResearchGate and Deutscher Bildungsserver (Line 106-107).

Point 3

The selection of 63 articles out of 1592 is not clear, it is not specified how many were eliminated with each criterion.

We added eliminated articles to each criterion in Figure 1.

Point 4

The presentation of results and the discussion are very interesting, but several points of the search strategy and the elimination of unreliable databases used need to be clarified.

Thank you for this comment. We have corrected the search strategy in Figure 1 and in the text (Line 106-107, Line 113).

I hope these comments will help you to improve your work.
